# Electrostatic catalysis of a click reaction in a microfluidic cell

Semih Sevim [1], Roger Sanchis-Gual[1], Carlos Franco[1], Albert C. Aragonès [2], Nadim Darwish [3], Donghoon Kim[1], Rosaria Anna Picca [4], Bradley J. Nelson [1], Eliseo Ruiz[5], Salvador Pané [1] ✉, Ismael Díez-Pérez [6] ✉ & Josep Puigmartí-Luis [2,7] ✉

Electric fields have been highlighted as a smart reagent in nature's enzymatic machinery, as they can directly trigger or accelerate chemical processes with stereo- and regio-specificity. In enzymatic catalysis, controlled mass transport of chemical species is also key in facilitating the availability of reactants in the active reaction site. However, recent progress in developing a clean catalysis that profits from oriented electric fields is limited to theoretical and experimental studies at the single molecule level, where both the control over mass transport and scalability cannot be tested. Here, we quantify the electrostatic catalysis of a prototypical Huisgen cycloaddition in a large-area electrode surface and directly compare its performance to the conventional Cu(I) catalysis. Our custom-built microfluidic cell enhances reagent transport towards the electrified reactive interface. This continuous-flow microfluidic electrostatic reactor is an example of an electric-field driven platform where clean large-scale electrostatic catalytic processes can be efficiently implemented and regulated.

Extensive theoretical work has long proposed electric fields as key driving forces of enzymatic catalysis[1]. Recognizing this importance, electric fields have been recently referred to as smart reagents in nature's catalytic machinery[2]. In nature, exceedingly large electric fields have been experimentally found to be electrostatically generated in confined spaces, e.g., in the enzymes' active sites[3], playing a crucial role in accelerating chemical reactions[4]. Theoretical efforts, such as computational predictions of the metal free Huisgen cycloaddition reaction under oriented external electric fields OEEF, have long predicted reaction rate and selectivity enhancements via applying OEEF along specific directions in relation to the reaction coordinate[5]. The technical challenges of generating oriented electric fields in a confined chemical reactor aligned with the reaction coordinate have hampered experimentation in this field. Nonetheless, some recent experimental efforts support the theoretical framework for electrostatic catalysis. Within the biological context, Fried et al. demonstrated that the enzyme ketosteroid isomerase exhibits a large electrostatic field within the active site, whose magnitude strongly correlates with the enzyme's catalytic efficiency[6]. Inspired by the latter, Aragonès et al. designed a nanoscale reactor in a tunnelling junction mimicking an enzyme's active site, where an electric field orients along the reaction coordinate of a confined non-redox Diels-Alder reaction, which resulted in a 5-fold reaction acceleration under moderate applied electric fields[7].

[1]Institute of Robotics and Intelligent Systems, ETH Zurich, Tannenstrasse 3, CH-8092 Zurich, Switzerland. [2]Departament de Ciència de Materials i Química Física, Institut de Química Teòrica i Computacional, University of Barcelona (UB), Marti i Franquès 1, 08028 Barcelona, Spain. [3]School of Molecular and Life Sciences, Curtin University, Bentley 6102 WA, Australia. [4]Chemistry Department, University of Bari "Aldo Moro", via E. Orabona 4, 70125 Bari, Italy. [5]Departament de Química Inorgànica i Orgànica, Institut de Química Teòrica i Computacional, University of Barcelona (UB), Diagonal 645, 08028 Barcelona, Spain. [6]Department of Chemistry, Faculty of Natural, Mathematical & Engineering Sciences, King's College London, Britannia House, 7 Trinity Street, London SE1 1DB, UK. [7]Institució Catalana de Recerca i Estudis Avançats (ICREA), Pg. Lluís Companys 23, 08010 Barcelona, Spain. ✉e-mail: vidalp@ethz.ch; ismael.diez_perez@kcl.ac.uk; josep.puigmarti@ub.edu

Exploiting the concept of electrostatic catalysis for practical applications inevitably goes through up scaling the process to allow mass production. While the latter has not been realized yet, pioneer efforts demonstrating electrostatic catalysis on a number of reactions on electrode surfaces[8–11] demonstrate the potential of this form of chemical catalysis for its use as a future clean, sustainable large-scale chemical processes. Towards this aim, several aspects need to be addressed: (1) quantifying the level of control over the reactant's orientation within the interfacial electric field, (2) quantifying the role of the electrical double layer (EDL), and (3), optimizing mass transport characteristics for the design of chemical engineering reactors. All the above requires to be benchmarked against the standard homogeneous catalysis pathway of the same reactions, as well as be optimized for low applied voltages to comply with low energy consumption. Likewise, the reaction portfolio is yet to be developed for other types of chemical reactions of practical relevance.

Here, we study the electrostatic catalysis of a prototypical non-redox Huisgen cycloaddition (click reaction), conventionally catalysed by copper(I) salts[12–18], over a large (centimetre scale) surface area. The reaction is effectively catalysed over a molecularly functionalized electrode surface exploiting voltage-controlled electric fields confined within an interfacial EDL (several nanometres in thickness) as the sole catalyst. The above catalytic processes are conducted under controlled mass transport conditions via a microfluidic channel, where turbulent flow and radial forces are avoided. This electrostatic continuous-flow microfluidic reactor displays superior catalytic performance for a click cycloaddition reaction of nearly 200% larger as compared to the classical Cu(I) chemical counterpart when electrostatically driven at small, applied voltages of 2 V under the same mass transport conditions. Our results bring several new key aspects over previous pioneer efforts[9,10]: (1) our work is an example of large-scale pure electrostatic catalysis in the absence of catalyst (neither homogeneous nor heterogeneous catalyst is present), (2) we show electrostatic catalytic effects at much lower applied potentials ($<1$ V) and quantify the effect against the conventional $Cu^+$ chemical catalysis, and (3), our microfluidic reactor allows us to perform these reactions under continuous (and laminar) flow conditions, which facilitates mass transport of reactants to the reactive interfacial EDL where the electrostatic catalysis occurs resulting in higher reaction yields. We believe this will engender new routes towards the implementation of a clean and metal free electrostatic catalysis in large scale applications.

## Results

### Electrostatic catalysis experiment in a microfluidic cell

To investigate the catalytic effect of an OEEF on a prototypical click reaction, we have chosen a standard azide-alkyne cycloaddition, which is typically attained with high yields using $Cu^+$-based catalyst[12–18], even on a functionalized solid support[19]. To this end, we functionalize a gold-coated glass surface with an azide-terminated molecule (**1** in Fig. 1a, left panels) via thiol-gold (S-Au) covalent chemistry[20] (see Supplementary Notes 1–3), and introduce a solubilized ferrocene (Fc) alkyne derivative (**2** in Fig. 1a, left panels) via a flow of acetonitrile solution of **2**. Upon reaction completion (Fig. 1a, right panel), the ferrocene exposed groups attached to the functionalized gold-coated electrodes allow for further quantification of the yield of the reaction via electrochemical cyclic voltammetry (CV)[21]. Note that the surface-confined azide groups force the reaction to occur within the EDL region that develops nearby the electrode surface in contact with the polar solvent (vide infra) under an applied voltage. To accomplish constant mass transport of the alkyne reactant (i.e., avoid surface mass depletion), the functionalized electrode was integrated into a microfluidic cell where the microscale channel is defined by two gold-coated glass electrodes separated by ca. 250 μm, which defines a reaction area of 5.7×1.2 cm² (Fig. 1b,c, Supplementary Figs. 1a−c, and Design of the microfluidic cell in Methods). Once assembled, this microfluidic cell

allows for introducing a continuous flow of a solution of **2** between the two electrodes while avoiding both turbulent flow conditions and poor control of the convective mass transport. A voltage applied between the two electrodes creates the OEEF at the electrodes/solvent interfaces via formation of an interfacial EDL (top-left panel in Fig. 1a).

In a typical experiment, a 0.1 M solution of ethynylferrocene in acetonitrile is injected between the two azide-functionalized gold electrodes of the microfluidic cell at a constant flow rate of 50 μL min⁻¹ for a reaction period of 30 minutes, while a voltage difference between 0.5–2 V is applied between the two parallel electrodes (see Microfluidic experimental procedure in Methods). The generated voltage-induced OEEF is expected to align the polar acetonitrile solvent molecules (dipole magnitude of 3.92 D) close to both electrified electrode surfaces forming EDL capacitors[22,23]. The utilized low voltage range (0.5–2 V), as compared to previous setups[10], ensures the experiments are done within the capacitance region (EDL) in acetonitrile, avoiding the generation of undesired faradaic processes at the electrified interface, including reductive thiol desorption of the azide moiety. The interfacial EDL formation results in the localization of the OEEF near the electrodes' surface, where the azide-alkyne cycloaddition reaction takes place. When the 30-minute reaction time concludes, the microfluidic cell is flushed with pure acetonitrile at a constant flow rate of 200 μL min⁻¹ for a period of 5 minutes to remove the unreacted ethynylferrocene. Successfully clicked ethynylferrocene reactant molecules to the azide-terminated electrode surface are then quantified via ex situ cyclic voltammetry of the redox ferrocene groups (see Supplementary Note 4). The integrated area (charge density) of the characteristic voltammetric peaks of ferrocene is proportional to the number of surface-bound ferrocene units, which is corroborated by surface XPS analysis (see Supplementary Figs. 2, 3), and used to determine the surface reaction yield[24]. We use this methodology to monitor the reaction yield under both electrostatic and chemical reaction conditions (Fig. 1a).

### OEEF magnitude effect on the electrostatic catalysis

We first evaluate the efficiency of the electrostatically catalysed azide-alkyne cycloaddition by comparing it to the same reaction under conventional $Cu^+$-based chemical catalysis conditions (Fig. 2a−c and Supplementary Fig. 4). A mixture of ethynylferrocene (0.1 M) and copper(I) iodide-triethyl phosphite (>20 mol% relative to the ethynylferrocene) in acetonitrile was injected between the two azide-terminated gold electrodes of the microfluidic channel using the same experimental conditions, namely, a constant flow rate of 50 μL min⁻¹ for a period of 30 minutes, and in the absence of OEEF (i.e., no voltage applied), leaving $Cu^+$ as the only catalysing agent for the click reaction (bottom-left panel in Fig. 1a). The resulting cyclic voltammetry study shows that the chemical $Cu^+$ catalysis results in a similar reaction yield as compared to the electrostatic catalysis (in the absence of $Cu^+$) performed at electric fields generated under very low (0.75 V) applied voltage (respectively, green and black curves in Fig. 2a). Ex-situ X-ray photoelectron spectroscopy (XPS) corroborates the existence of iron on the azide-terminated electrode surfaces, prepared under both electrostatic and $Cu^+$-based chemical catalysis (Supplementary Table 1). The XPS data also indicates that the electrostatic catalysis yields much cleaner surfaces than the $Cu^+$-based catalysis, the latter leaving residues from the employed $Cu^+$ salt such as adsorbed iodide ions on the surface of the Au electrode (Supplementary Table 1). Iodine contamination is also observed in consecutive CV cycles of samples prepared under $Cu^+$-based catalysis as an irreversible second oxidation peak corresponding to the $[I_3]^- \rightarrow I_2$ reaction[25] (Supplementary Fig. 5). The absence of copper signal in the spectroscopic data suggests that it is eliminated from the substrate during the washing step.

Figure 2a and b show that the magnitude of the applied OEEF provides a fine control knob for the reaction yield. Different voltages ranging from 0.5 to 2 V were used to electrostatically catalyse the click

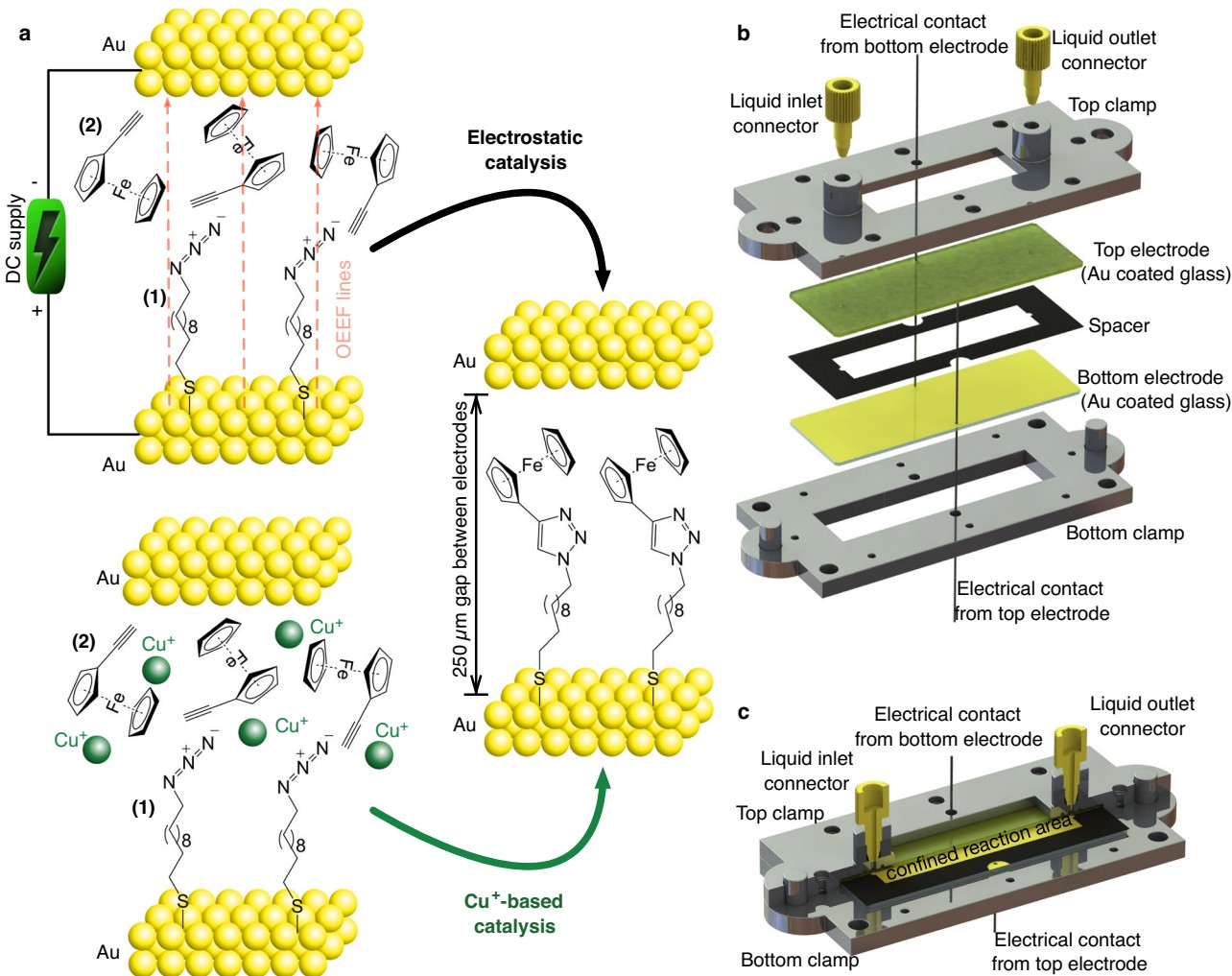

**Fig. 1 | Catalysis of a click reaction in a microfluidic cell. a** Schematic representation of oriented external electric-field (OEEF) in an electrostatic catalysis (top panel), and copper (Cu⁺)-catalyzed click cycloaddition (bottom panel) in a confined microfluidic channel between two gold electrodes. The azide moiety (**1**) is immobilized on the gold surface via thiol-gold chemistry, and the ferrocene alkyne derivative (**2**) is flowed continuously to avoid its depletion on the functionalized gold surface. **b, c** Schematic drawing showing parts of the microfluidic cell prior to its assembly (in panel **b**) and the assembled microfluidic cell with a section-cut in the top part to demonstrate the confined reaction area formed between two gold electrodes using a spacer (in panel **c**).

reaction. CV results show net electrostatic catalysis for applied voltages as low as 0.75 V (Fig. 2a, b). Electrostatic catalysis performed at lower voltages than 0.5 V resulted in CVs with no evident redox signal of ferrocene (grey curve in Fig. 2a) resembling those of bare and/or azide-terminated gold substrate (respectively orange and purple curves in Supplementary Fig. 6). The reaction yield increased by 22% and 197% at applied voltages of 1.5 V and ≥2 V, respectively, over the values obtained for the Cu⁺-based catalysis (Fig. 2a, b), indicating superior catalytic performance in the electrostatic context. Specifically in the case of 2 V applied voltages, the integrated charge density reaches a charge density of 0.4 C m⁻², which corresponds to a ferrocene surface coverage of $2.47 \pm 0.13$ nm⁻², a value very close to the theoretically calculated fully covered ferrocene layer (2.7 nm⁻²)[26,27]. Additionally, the obtained linear relation between the scan rate and peak current density evidences that the reversible redox couple comes from surface-bound redox species (Fig. 2c)[24], as expected from chemically-bound ferrocene on the Au electrode surface. It is important to note that all the experiments comparing reaction yields were performed using azide-terminated electrodes prepared in a single batch (i.e., same metallization and functionalization batches). Due to possible batch-to-batch differences in the packing density of the surface functional groups, we observed slight variations regarding the

shapes and areas of the voltammetric peaks as a result of disordered arrangements or supramolecular interactions between neighbouring cyclopentadienyl groups[28,29]. Despite the latter, the voltage-dependent catalytic effect was consistent across all batches (see results for a highly dense ferrocene layer under different OEEFs, blue and black curves in Supplementary Fig. 6).

## Mass-transport effect on the electrostatic catalysis

We conducted the electrostatic catalysis experiment under stagnant conditions, i.e., stopped-flow (Fig. 2d), and compared it to the in-flow reaction shown in previous section. We used square-wave voltammetry (SWV) in this case to minimize the contribution of the capacitive current and increase the sensitivity (i.e., peak integrability) of the low voltammetric signal for the stop-flow experiment[30]. The reaction yield for the in-flow conditions was 87% larger than the stop-flow under the same applied OEEF of 1 V (solid and dashed black curves in Fig. 2d). We observe that the enhanced mass-transport conditions under continuous flow improves final reactions yields across different applied OEEFs, which shows the key role of mass-transport and that the reaction yield can be tailored both with the OEEF magnitude (i.e., applied voltage) and controlled mass transport (i.e., flow conditions).

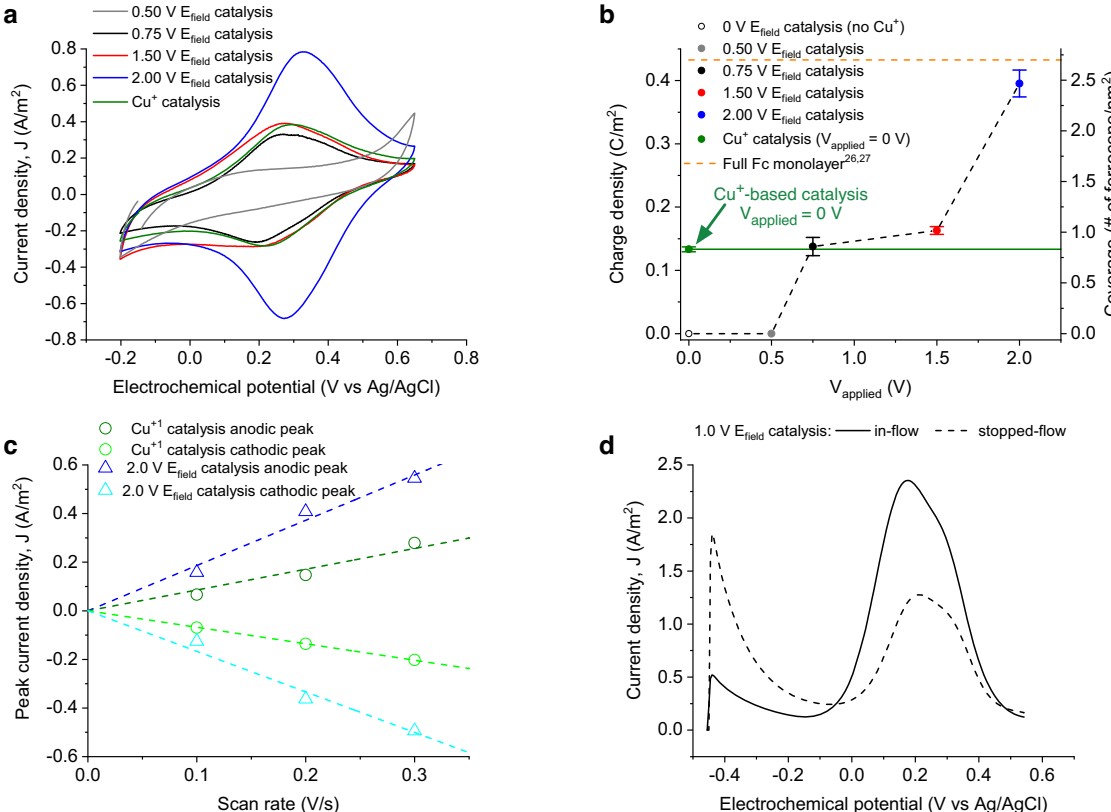

**Fig. 2 | Performance of the electrostatic catalysis. a** Cyclic voltammetry (CV) data showing effect of the applied voltage in the electrostatic catalysis, e.g., 0.5 V (grey), 0.75 V (black), 1.5 V (red), 2 V (blue), and the comparison with Cu⁺-based chemical catalysis (green). The two major peaks correspond to the oxidation-reduction of the ferrocene moieties attached to azide-terminated electrode via azide-alkyne cycloaddition. The CVs in a are obtained at a scan rate (SR) of 0.2 V s⁻¹. **b** Integrated charge densities from the anodic (positive) peaks of the CVs for OEEF catalysis at 0.50 V (grey dot) 0.75 V (black dot), 1.5 V (red dot) and 2 V (blue dot), and for Cu⁺-based catalysis at 0 V (green dot). Open circle is the control experiment performed under non-catalytic conditions (i.e., $V_{applied}$ = 0 V

and absence of Cu⁺) and the dashed orange line indicates the ideal fully covered ferrocene monolayer from the literature[26,27]. Error bars represent the standard deviation of results obtained from n experiments (3≤ $n$ ≤6) at each applied voltage. **c** Scan rate dependency (see also Supplementary Fig. 4) of the anodic and cathodic peak current densities obtained from the CV analysis of samples prepared using electrostatic catalysis at 2 V (triangles) and Cu⁺-based chemical catalysis (circles) together with the corresponding linear fits (colour coded dashed lines). **d** Square-wave voltammetry (SWV) of electrostatically catalysed (at 1 V applied voltage) surfaces under continuous- (solid line) and stop- (dashed line) flows in the microfluidic cell.

## Kinetics of the electrostatic catalysis

To assess the conversion rates for electrostatic catalysis and compare it with the conventional Cu⁺ catalysis, we performed time-dependent experiments of both electrostatic (at +2 V) and chemical Cu⁺-based catalysis (Fig. 3a–b). As seen in Fig. 3a and c, after only 10 min reaction times, the OEEF-induced catalysis yields considerable larger ferrocene (Fc) redox charge densities (0.15 C m⁻²), corresponding to a Fc converge of 0.94 nm⁻², and reaches almost full surface saturation, ~2.7 nm⁻²[226,27], after reaction times of ≤60 min (orange dashed line in Fig. 3c). Note that each data point in the graph showing the time dependency of the azide-alkyne cycloaddition reaction was obtained from different sample preparations, corroborating the robustness of the methodology. In the case of the Cu⁺ catalysis, we obtain a much slower reaction kinetics, with unappreciable product formation within the first 20 min of reaction time (red and black CVs in Fig. 3b and three first green dots in 3c). Near full Fc coverage (2.45 nm⁻²) is not achieved until 120 min of reaction time. The average slopes corresponding to the experimental reaction rates, $2.03 \times 10^{-2}$ and $5.24 \times 10^{-2}$ Fc nm⁻² min⁻¹ respectively for Cu⁺ and electrostatic catalysis cases (green and blue dashed lines in Fig. 3c), yield more than twice (2.58 times) as much conversion rates for the electrostatic case, thus corroborating the superior performance of the electric field catalysis.

## Quantification of the EDL-based OEEFs

In view of the above results, we attempted to quantify the magnitudes of the OEEFs generated within the EDL at the electrode/liquid interface to see whether the values are consistent with the observed electrostatic chemical catalysis. To this aim, we use voltage pulses followed by chronoamperometry (CA) measurements to determine the charge associated with the charge/discharge process of the voltage-induced interfacial EDL capacitor[31]. This was done across different electrochemical cell arrangements as a comparative study (see Supplementary Note 4 and Supplementary Fig. 7). A step potential equivalent to the voltage ($V_{applied}$) used in the OEEF-based catalytic experiment is applied between the two parallel electrodes in pure acetonitrile, and the resulting current transient is recorded. The measurements are repeated for consecutive charging and discharging cycles (Fig. 4a). The transient showed fast current decay to zero level characteristic of the EDL charge/discharge capacitive process, which also shows the absence of faradaic current (parasitic electrochemical processes) within the employed voltage range. Note that open circuit potential ($V_{OCP}$) were chosen to be the minimum (and initial) potential in the CA measurements in order to attain high reproducibility and less hysteresis[31,32] (see Supplementary Note 4). The transient decay curves were fitted to an exponential function using an $RC$ circuit as a model to obtain the time constant of the EDL-based capacitor ($C$) (Fig. 4b) and,

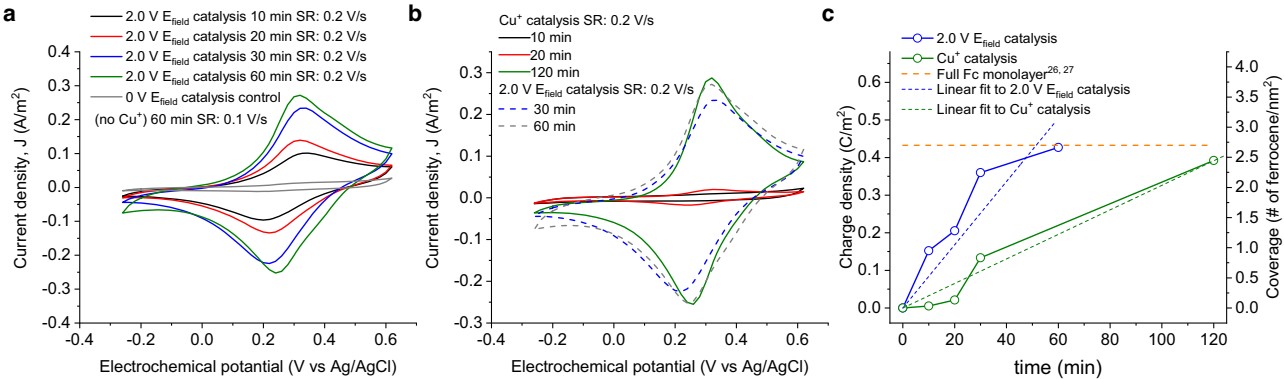

**Fig. 3 | Electrostatic catalysis kinetics. a** CVs obtained from the samples prepared under electrostatic catalysis at a bias voltage of 2 V for 10 min (black), 20 min (red), 30 min (blue), and 60 min (green) and compared to the control experiment in the absence of both electric field ($V_{applied}$ = 0 V) and Cu$^+$ catalysis (grey). **b** CVs obtained from the samples prepared under chemical Cu$^+$ catalysis for 10 min (black), 20 min (red), 120 min (green) and compared to the samples prepared with electrostatic catalysis for 30 min (dashed blue) and 60 min (dashed grey). In a and b, SR represents the scan rate. **c** Reaction yields expressed as both ferrocene (Fc) redox charge density (left Y-axis) and Fc surface coverage (right Y-axis) as a function of time for the electrostatic catalysis conducted at 2 V bias voltage (blue solid line with circles) and Cu$^+$ catalysis (green solid line with circles), and corresponding linear fits (color coded dashed lines). The dashed orange line indicates the ideal fully covered ferrocene monolayer from the literature[26,27].

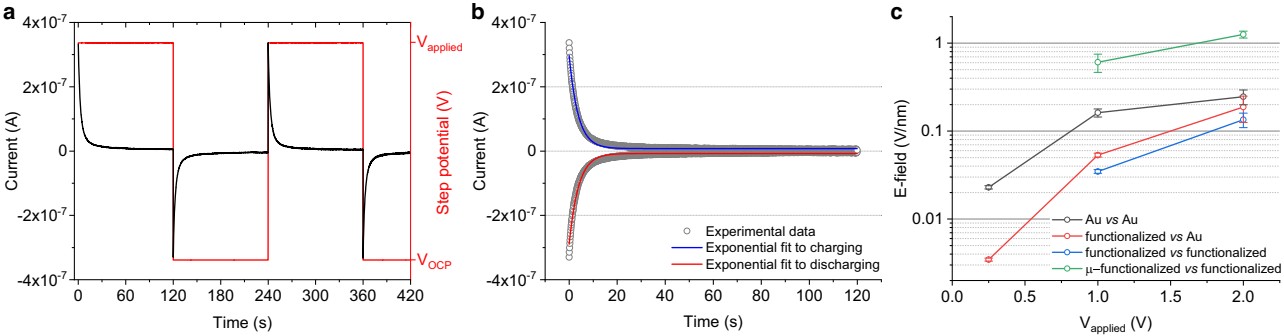

**Fig. 4 | Characterization of the EDL-based OEEFs using an *RC* circuit model. a** Representative chronoamperometry (CA) recordings showing charging and discharging currents (black) in consecutive applied step voltages (red) between the parallel electrodes immersed in pure acetonitrile. **b** Exponential fits (blue and red lines) to the charging and discharging currents (grey circles) using an *RC* circuit as a model. **c** Calculated electric fields from the *RC* fittings corresponding to different electrode configurations: in a beaker containing pure acetonitrile and (1) two parallel bare gold electrodes (Au vs Au, black line), (2) an azide-terminated gold electrode against a bare gold electrode (functionalized vs Au, red line), (3) two parallel azide-terminated electrodes (functionalized vs functionalized, blue line), and (4), two parallel azide-terminated electrodes confined in the microfluidic cell (μ-functionalized vs functionalized, green line). Error bars represent the standard deviation for number of samples, $n \geq 3$. See also Supplementary Fig. 7 for detailed schematic representations of electrochemical cell configurations used in CA measurements.

hence, the associated charge (*Q*) and corresponding OEEF (*E*) across the EDL (Table 1).

We investigated the magnitudes of the OEEFs in the same voltage range (from 0.25 V to 2.0 V) employed in the electrostatic catalysis experiments (see black, red and blue lines in Fig. 4c). The CA experiments were done in both a conventional electrochemical cell and the same microfluidic cell (see details in Supplementary Note 4). The former electrochemical setup provides fine control on the applied electrochemical potentials to both electrodes' interfaces and serves as a benchmark for the quantification of the EDL charging/discharging process. The experiments performed in the conventional electrochemical cell using acetonitrile and bare Au electrodes (Supplementary Fig. 7a) results in interfacial OEEFs of up to 0.25 V nm$^{-1}$ at 2 V (Table 1). The additional functionalization of either one or the two electrodes acted as additional capacitance elements combined in series, which contributes towards charge shielding, hence decreasing the EDL-based OEEF generated under the same 2 V voltage down to 0.19 and 0.13 V nm$^{-1}$ respectively (Supplementary Figs. 7b, c and Supplementary Table 1). We then compare the above OEEFs to those equivalently generated in the microfluidic channel. To this aim, we filled our microfluidic cell with pure acetonitrile and applied voltage steps of 1 and 2 V to the two parallel electrodes, both decorated with the azide-terminated compound (Supplementary Fig. 7d). The resultant OEEF magnitudes appear to be significantly larger in the microfluidic channel, $0.61 \pm 0.14$ V nm$^{-1}$ and $1.26 \pm 0.11$ V nm$^{-1}$, respectively for 1 V and 2 V (see Table 1 and green line in Fig. 4c). While we do not have a clear picture for this phenomenon, we hypothesize that the confinement of the polar medium within the microfluidic channel yields a more effective alignment of the polar solvent molecules under the applied electric field resulting in more compact EDL and corresponding higher EDL-based OEEF. OEEFs in the order of 1 V nm$^{-1}$ have been proven to induce chemical catalysis in both experimental and computational contexts[7,22,33–35].

## OEEF polarity effect on the electrostatic catalysis

The OEEF polarity is a crucial parameter in the mechanisms of electrostatic catalysis, since, in a pure electrostatic catalysis process, opposed directions of the applied OEEF yield completely different reaction yields[22]. Figure 5c shows the reaction yield upon OEEF polarity inversion drops around 73% as compared to the positive polarity, even at moderates applied OEEFs of 0.75 V. The effect is more drastic at higher negative bias voltages (see Supplementary

**Table 1 | Calculation of the EDL-based OEEF magnitudes**

| Configurations | | $V_{applied}$ (V) | C (μF) | Q (μC) | E (V nm$^{-1}$) |
|---|---|---|---|---|---|
| Conventional electrochemical cell* | 1st configuration: (Au vs Au) | 0.25 | 28.6 ± 1.3 | 7.1 ± 0.3 | $2.3 \times 10^{-2} \pm 1.0 \times 10^{-3}$ |
| | | 1.00 | 50.3 ± 5.3 | 50.3 ± 5.3 | $1.6 \times 10^{-1} \pm 1.7 \times 10^{-2}$ |
| | | 2.00 | 38.3 ± 7.3 | 76.5 ± 14.6 | $2.5 \times 10^{-1} \pm 4.7 \times 10^{-2}$ |
| | 2nd configuration: (Functionalized vs Au) | 0.25 | 4.3 ± 0.2 | 1.1 ± 0.0 | $3.5 \times 10^{-3} \pm 1.3 \times 10^{-4}$ |
| | | 1.00 | 16.6 ± 0.9 | 16.6 ± 0.9 | $5.3 \times 10^{-2} \pm 2.8 \times 10^{-3}$ |
| | | 2.00 | 29.2 ± 9.7 | 58.5 ± 19.4 | $1.9 \times 10^{-1} \pm 6.2 \times 10^{-2}$ |
| | 3rd configuration: (Functionalized vs functionalized) | 1.00 | 10.8 ± 0.5 | 10.8 ± 0.5 | $3.5 \times 10^{-2} \pm 1.7 \times 10^{-3}$ |
| | | 2.00 | 20.9 ± 3.9 | 41.9 ± 7.8 | $1.3 \times 10^{-1} \pm 2.5 \times 10^{-2}$ |
| Microfluidic cell# | 4th configuration: (μ-functionalized vs functionalized) | 1.00 | 138.1 ± 32.4 | 138.1 ± 32.4 | $6.1 \times 10^{-1} \pm 1.4 \times 10^{-1}$ |
| | | 2.00 | 132.5 ± 22.7 | 265.0 ± 45.3 | $1.3 \times 10^{0} \pm 1.1 \times 10^{-1}$ |

OEFF magnitude (mean ± std, for number of samples, $n \geq 3$) calculated by using an RC circuit model for constant voltages ranging between 0.25 V and 2 V in the different employed electrochemical configurations.
*Spacing between parallel electrodes is 7 mm, #spacing between parallel electrodes is 250 μm.

Fig. 9). We rationalize the impact of the OEEF polarity on the reaction kinetics as 2-fold; (i) the magnitude of the DFT calculated reactants' dipole moments (~1.5 D, see Supplementary Fig. 8 and Supplementary Table 2) times the electric field generated within the microfluidic cell (~1 V nm$^{-1}$, see Table 1) yield interaction energies, $U = -|\overrightarrow{OEEF}|*|\overrightarrow{\mu}_{reactants}|$, well above $kT$ values, which results in the reactants being effectively oriented along their dipoles under the working OEEF. The resulting reactants' orientation for a positive OEEF polarity offers a more favorable approach for the formation of the transition state (TS) between the Au-anchored terminal azyde and the free Fc-alkyne (Fig. 5a), leading to the 1,5-isomer product. Moreover, the reaction path leading to the 1,5-isomer scores the larger dipole moment increment (see Supplementary Table 2), making it the most kinetically favorable one ($\triangle\triangle G^{\neq} = -|\overrightarrow{OEEF}|*(|\overrightarrow{\mu}_{TS}| - |\overrightarrow{\mu}_{Reactants}|)$), assuming similar dipole values for both TS and products given their structural homology (Supplementary Table 2 and Fig. 5a). And (ii), both OEEF polarities would a priori effectively stabilize the charge-separated state of one of the resonance contributors in the TS (Fig. 5b), thus reducing the energy barrier[2,4,7,22,34–38] and accelerating the reaction.

**Solvent polarity effect on the electrostatic catalysis**
Towards supporting the above EDL-based electrostatic catalysis, we turned our attention to the polarity of the reaction medium (Supplementary Note 6). Considering the EDL argument as the origin of the interfacial OEEF, a less polar solvent should hamper the OEEF localization near the electrode surfaces due to its much less efficient shielding of surface charges, leading to a thicker EDL spanning a larger distance from the electrode surface which results in lower OEEF magnitudes for the same applied voltages (Fig. 5d). Due to solubility constrains, we could not use a non-polar solvent and the comparison was done against toluene whose relative polarity versus water is 0.1, roughly 4 times less than acetonitrile scoring 0.46[39]. Using the standard chemical Cu$^+$ catalysis as the benchmark for comparison of the OEEF-based catalysis across both solvents, we observe that the OEEF-based catalysis at 1.5 V voltage in toluene displays a ~30% lower reaction yield than the conventional Cu$^+$ catalysis in the same solvent (solid red and green curves in Fig. 5e), whereas the OEEF-based catalysis at the same voltage in acetonitrile resulted in a higher observed yield when compared the its homologous Cu$^+$ catalyzed reaction in the same solvent (Fig. 2a–b). These results demonstrate the EDL nature of the acting localized OEEF at the reactive electrode/solvent interface inducing the electrostatic catalytic processes[40,41].

In summary, we have presented the first example of a microfluidic-based electrostatic approach for large scale electric-field driven catalysis under controlled mass transport. We demonstrated that the applied OEEF between the two electrodes confining the microfluidic channel can be used as a knob to regulate the yield of a prototypical Huisgen cycloaddition reaction generated at a large-area electrode/liquid interface. The correlation of the reaction yield with both the magnitude and polarity of the applied OEEF evidence the electrostatic nature of the catalysis. We also showed the key role of EDL in confining OEEF at the reactive electrode/liquid interface using solvents of different polarities, which results in different EDL thicknesses under the same applied voltages, hence regulating the final OEEF magnitude. The quantification of EDL-based OEEFs proves that the confinement of electrodes in a microfluidic reactor under exceptional mass transport conditions facilitates high interfacial OEEFs, thus achieving more efficient electric-field driven catalysis. We anticipate that our electrostatic continuous-flow microfluidic reactor will serve as a platform to open new routes to electric-field-driven chemical synthesis towards a clean, more sustainable chemical synthesis.

## Methods
### Design of the microfluidic cell
The microfluidic cell was designed as two separate parts (i.e., the top clamp and the bottom clamp, Supplementary Fig. 1a) using a 3D computer-aided design (CAD) software (SOLIDWORKS 2018) to mechanically clamp a spacer – e.g., ca. 250 μm-thick acrylonitrile butadiene rubber (NBR) sheet or ca. 100 μm-thick polytetrafluorethylene (PTFE) – between two gold-coated glass slides (i.e., electrodes), hence forming a reaction area confined between parallel electrodes (Supplementary Fig. 1b). The top and bottom parts of the microfluidic cell were machined from aluminum. Specifically, the top part includes input/output ports for microfluidic connectors (10-32 Coned for 1/16" OD, IDEX Health & Science, LLC, USA) to connect PTFE tubing (1/16" OD, IDEX Health & Science, LLC, USA) and hence enable continuous-flow operations. Moreover, both the top and bottom parts have extra holes that match with the ones in the gold coated glass slides, and that are used to make the electrical contact to the electrode surfaces (Supplementary Figs. 1a–d). Once the microfluidic cell is assembled, it enables for not only the continuous-flow (with a syringe pump, neMESYS 290 N, CETONI GmbH, Germany) of reagent solution between the functionalized working and counter gold electrodes but also the application of a bias voltage with a DC power supply (RND 320-KA3005D, RND Lab, Distrelec Group AG, Switzerland) to generate an oriented external electric field (OEEF) between these parallel gold electrodes.

### Microfluidic experimental procedure using acetonitrile as a solvent (high polar medium case)
Note that all reagent solutions were filtered prior to injection of microfluidic device to avoid any precipitation in solution.

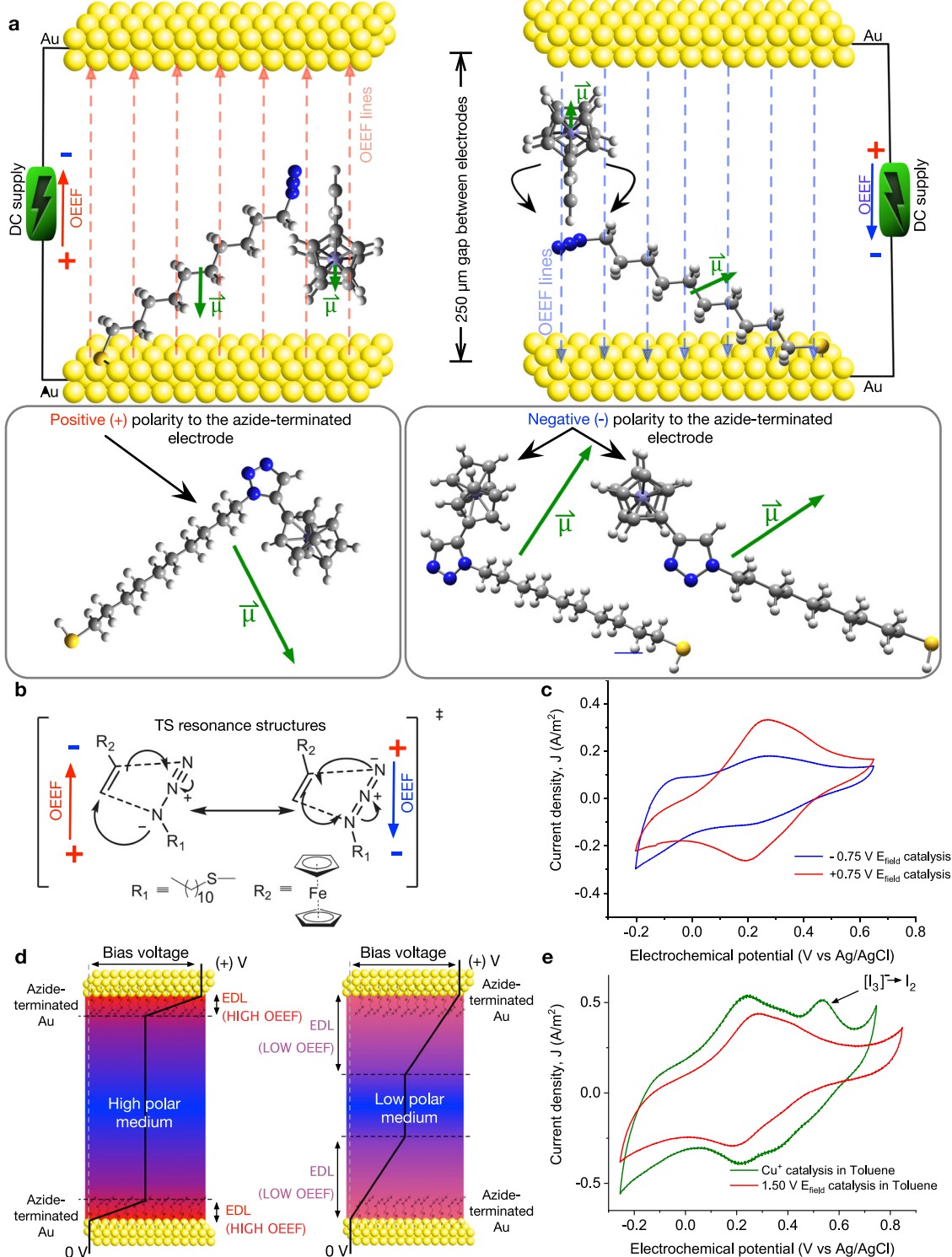

**Fig. 5 | Effect of OEEF and medium polarity in the electrostatic catalysis.**
**a** Schematic representation of the OEEF lines and the alignment of the polar ethynylferrocene and azide according to their dipole moments with (left) positive (+) and (right) negative (−) applied voltage with respect to the azide-terminated working electrode. Two plausible clicked products of reaction are shown depending on polarity of the working electrode. **b** Most likely transition state (TS) resonance structures stabilized under opposite OEEF polarities (e.g., positive (+) and negative (−) applied voltages in left and right structures, respectively). **c** Cyclic

voltammograms of electrostatically catalysed chemical reactions using opposite voltage polarities of −0.75 V (blue) and +0.75 V (red). **d** Schematic diagram showing EDL formation and corresponding voltage drop in high polar (acetonitrile, left) and low polar solvents (toluene, right) respectively. **e** Cyclic voltammograms of samples prepared in low polar medium (toluene) under Cu⁺-based chemical catalysis (solid green line) and electrostatic catalysis at a voltage of 1.5 V (solid red line). Cyclic voltammograms presented in c and e were obtained at scan rates of 0.2 V s⁻¹ and 0.3 V s⁻¹, respectively.

Electrostatic catalysis under continuous-flow conditions: (i) The microfluidic cell was assembled by clamping a 250 µm-thick NBR spacer between the gold-coated glass slides (i.e., the azide-terminated working and counter electrode) to define a confined reaction area (57 mm × 12 mm) between the two parallel electrodes (separation distance ca. 250 µm) (Supplementary Fig. 1c). (ii) The electrical contact for the top (or bottom) electrode was established with a conductive wire going through the concentric holes on the bottom (or top) clamp, bottom (or top) glass slide and spacer until touching to the top (or bottom) electrode surface (Supplementary Fig. 1a and c). (iii) Prior to introducing the reagent solution (i.e., 0.1 M ethynylferrocene in acetonitrile), the microfluidic cell was flushed with solvent (i.e., pure acetonitrile) with a flow rate of 200 µL min$^{-1}$ for 5 minutes. (iv) Then, 0.1 M ethynylferrocene in acetonitrile was injected in the microfluidic cell (at a flow rate of 200 µL min$^{-1}$ for ca. 2 minutes) to remove the solvent and quickly fill the reaction chamber with the reagent solution. (v) Once the reaction chamber was completely filled with the reagent solution, the flow rate was decreased to 50 µL min$^{-1}$ and a constant DC bias voltage was applied to the parallel electrodes for a period of 30 minutes. Note that we also performed experiments with different constant applied voltages ranging between 0 V–2 V. (vi) After the 30 minutes reaction time under continuous-flow conditions, the applied voltage was stopped, and pure acetonitrile was injected (200 µL min$^{-1}$ for 5 minutes) in the microfluidic cell to remove the excess ethynylferrocene solution and clean the electrode surfaces to prevent any physiosorbed molecules or surplus reactants. (vii) Finally, the microfluidic cell was dissembled, and the working electrodes were characterized in a separate electrochemical cell to a posteriori interrogate the yield of click reaction (see Supplementary Note 4 for more detail).

Electrostatic catalysis under stopped-flow conditions: The same experimental procedure, described for Electrostatic catalysis under continuous-flow condition, was followed except for step (v). More specifically, while applying the bias voltage to the parallel electrodes the flow was stopped (i.e., flow rate was decreased to 0 instead of 50 µL min$^{-1}$).

Cu$^+$-based chemical catalysis under continuous-flow conditions: The same experimental procedure, described for Electrostatic catalysis under continuous-flow condition, was repeated with a reagent solution containing Cu$^+$ as catalyzing agent – i.e., Copper(I) iodide-triethyl phosphite (>20 mol% relative to the ethynylferrocene) was added to 0.1 M ethynylferrocene solution in acetonitrile – and no bias voltage was applied during the reaction step, (step (v)).

**Computational details**

DFT calculations were performed by using the Gaussian 16 program[42] with exchange-correlation functional B3LYP[43] together with TZVP basis set[44]. To simulate the solvent effect a CPCM model[45] was employed. DFT calculations show that the solvent dielectric constant (toluene 2.38 and acetonitrile 37.5) increases the change in the dipole moment Δµ (see Supplementary Table 2) between reactants and products in the click reaction. Consequently, an external electric field favors the reaction in solvents with a higher dielectric constant. More detailed information on materials, additional experimental and characterization methods, as well as theoretical calculations could be found in Supplementary Information.pdf file.

## Data availability

All data needed to evaluate the conclusions in this work are present in the main text and in the Supplementary Information. Supplementary Information.pdf file includes Supplementary Figs. 1–9; Supplementary Tables 1–2; Supplementary Notes 1-6 regarding more detailed information on materials, additional experiments, characterization methods, and theoretical calculations; and Supplementary References. Additional data related to this paper may be requested from corresponding authors.

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

## Acknowledgements

This work is supported by European Union's Horizon Europe Research and Innovation Programme under EVA project (GA No: 101047081, JPL, SP); European Research Council Consolidator Grant Fields4CAT ERC –2019-CoG 772391 (IDP); Swiss National Science Foundation 200021_181988 (JPL); MCIN/ AEI /10.13039/501100011033 PID2020-116612RB-C33 (JPL); 2021 SGR 00270 (JPL); PID2021-122464NB-I00 (ER); CEX2021-001202-M (ER, JPL); ICREA Academia (ER). Partial support from SNSF-Sinergia project no:198643 (BJN) is also acknowledged. S.P. acknowledges the Swiss State Secretariat for Education, Research and Innovation (SERI). IDP acknowledges financial support from the UKRI Biotechnology and Biological Sciences Research Council (BBSRC) under the grant agreement BB/X002810/1.

## Author contributions

S.P., I.D.P. and J.P.L. conceived the idea. S.S. designed the microfluidic cell, planned and performed all experiments, analyzed and interpreted the data. N.D. helped conceive the idea and with the data analysis. S.S., R.S.G. and C.F. performed the electrochemical characterizations. A.C.A. and D.K. helped with some of the experiments and the electrochemical characterizations. R.A.P. performed the XPS measurements. B.J.N. helped with the data analysis. E.R. performed the DFT calculations. S.S. and I.D.P. wrote the manuscript. B.J.N., S.P. and J.P.L. contributed to writing the final version of the manuscript.

## Competing interests

The authors declare no competing interests.
