## [Peer Review File · Nature Communications]

Electrostatic catalysis of a click reaction in a microfluidic cellReviewers' Comments:

Reviewer #1:

Remarks to the Author:

For the paper, titled "electrostatic catalysis of a click reaction in a microfluidic cell", Sevim et al. quantified the role of the electrical double layer on the electrostatic catalysis of a prototypical Huisgen cycloaddition with controllable mass transport. They designed a microfluidic cell, which could enhance reagent transport towards the electrified reactive interface and avoid both turbulent flow conditions and poor control of the convective mass transport. Their designed microfluidic reactor presents superior catalytic performance for cycloaddition reaction by 200% larger at a small, applied voltages of 2 V than that over a conventional Cu(I) chemical counterpart.

Here are some comments for the paper to address before the consideration of publications.

1. What's the curvature of Au electrode? Fig 1 shows flat slab of Au. This is very confusing. How the Au electrode without any curvature can achieve such a high electric field of ~ 1 V/nm at a very small, applied potential?
2. Did authors explore a synergistic effect of combining both electrostatic reactions and Cu(I) catalysts? Since Cu is positively charged, will that further enhance the local Efield?
3. The calculated OEEF magnitudes need a further validation from experiments, e.g., stark vibrational frequency.
4. A lack of information about the selectivity and conversion rates of the reaction when using OEEF. Will that be different as compared to the conventional Cu(I) catalyst?
5. The author's description on page 8 only focuses on the positive potential. Could the authors elaborate on this observation at negative potentials? In general, if authors applied the field correctly, negative potential should show opposite trend for the reaction performance as compared to that under a positive electric field.
6. On page 11, while discussing Fig. 4b, you attribute the effects of OEEF polarity to the stabilization of the charge-separated state in the transition state. Besides the provided references, are there any additional experimental or theoretical evidence that you can present to substantiate this stabilization effect in the transition state of the azide-alkyne cycloaddition reaction?
7. More in-depth explanations of DFT calculations are needed, such as the model itself, how they added external field, how the TS was stabilized by the field, does the model solely incorporate azide-terminated molecules, or is gold (Au) also included as a catalyst? In addition, would variations of the reaction species potentially impact the change in dipole moment?
8. The manuscript would benefit from providing more detail on experimental reproducibility. Were the experiments performed multiple times, and were the results consistent? If the authors did multiple times, can they add the error bar for the results?

Reviewer #2:

Remarks to the Author:

Review of Sevim et al.

In the attached manuscript, Sevim and colleagues present data regarding a microfluidic device that can perform electrostatic catalysis through a static field generated by a potential difference between two electrodes. They demonstrate catalysis of an azide-alkyne cycloaddition reaction without the exogenous need for Cu catalysts. On the whole, the work appears to be a nice contribution to the discipline of chemical catalysis, demonstrating how device engineering and electrostatics can be leveraged together to build out a new way to catalyze chemical reactions. This reviewer has not been active in this precise field for some time, and must confess that superficially the work appears similar to the decade-old pioneering studies by Gorin et al. It may behoove the authors to be clearer in their introduction and discussion what the differences/improvements are between their system and previously-reported ones. On the whole, the work is well argued and evidenced, and seems potentially worthy of publication.

The following are specific comments, major followed by minor:

1. The authors refer in many places to their reaction occurring in a confined area/space. I'm not sure if I agree with this assessment. Generally, the term "confined" when used by supramolecular/polymer/colloid chemists refers to scenarios when movement is restricted to molecular length scales. The author's device is 250 μm thick, which is still nearly one million times larger than molecular dimensions. It seems to me that the molecule's diffusivity properties in the device would be quite similar to a bulk phase.

2. The authors are keen to make the claim that their device catalyzes the cycloaddition reaction, and based on the data in Fig. 2a, it seems that this is likely the case; however, catalysis has to do with the acceleration of rates, and the authors do not directly report this because (as far as I can tell) all reactions were conducted for 30 min without any direct measurement of kinetics. Minimally, the authors would need to report conversions as a function of time to obtain rates, and then OEEF-induced catalysis could then be more solidly supported based on rate accelerations.

3. Reaction progress is monitored through the current density associated with the redox couple of Fc/Fc^+ , and using this assay the authors can conclude that higher fields result in more surface-attachment of ferrocene. It seems like it would behoove the authors if this measure of yield could be converted to a more conventional measurement (such as % of azides on surface that got converted). This could be done by establishing a calibration curve constructed by pre-synthesizing the ferrocene-*triazole*-thiol in solution, putting it down on the Au surface, and measuring the current density when a known amount of ferrocene is down on the surface

4. The following sentence in the manuscript seems quite incorrect: "The OEEF polarity impact on the reaction is 2-fold; (i) on the kinetics; it increases reaction yield of the azide-alkyne cycloaddition by facilitating the proper alignment of the free ethynylferrocene reactant in solution with respect to the surface-bound azide group³⁶ (Fig. 4a), and (ii) on the thermodynamics; it stabilizes the charge separated state of one of the resonance contributors in the transition state (TS) (Fig. 4b), thus reducing the energy barrier."

Firstly, kinetics is about rates – not yields. Returning to my point #2, the authors ought to make more direct measurements on rates to make claims about kinetics. The second part of the sentence does not seem correct either. Thermodynamics governs which species prevail at equilibrium. Stabilizing the charge configuration of a transition state would accelerate a reaction and impact kinetics, not thermodynamics.

A. In connection to this point, since the azide-alkyne cycloaddition is effectively irreversible, it is under kinetic control, not thermodynamic control. So what matters in controlling the rate is the change in dipole moment upon going to the transition state: $\Delta G^{\ddagger} = \vec{E} \cdot \Delta \vec{\mu}_{R \rightarrow TS}$.

5. The last part of the paper shows some interesting experiments looking at the effect of solvent polarity and showing that the effect on yield is consistent with the predicted decrease in the EDL magnitude. Could this be compared to classic electrostatic theory? Also, would the authors predict a further acceleration if the linker of the alkane thiol were shorter? Presumably 11 carbons would also place the azide further away from the strongest portion of the EDL?

1. "Following nature's inspiration, the above catalytic processes is conducted under controlled mass transport conditions via a confined microfluidic channel, where turbulent flow and radial forces are avoided." If by Nature, the authors are referring to biology, then this sentence does not make sense because biology does not use microfluidic channels to promote catalysis (as far as I know)

2. "We believe this is a significant jump towards the implementation of a clean electrostatic catalysis in chemical industry applications." Seems like rather editorialized language

3. "Upon reaction is completed." Grammar

4. Showing the absence of reaction in the absence of field in Figure 2 would be helpful (though it looks

like the authors did this experiment and reported it in Figure S6)

5. In Table 1, the authors calculate the electric field based on a RC model. It would be good to show the equation of how this was derived based on the observed quantities. Also would be good to have error bars in Table 1.

6. Related to the previous point, if I'm not mistaken there is pretty good analytical theory for predicting EDL fields like Chapman-Gouy? Couldn't the authors compare their values to that from these classic models?

7. The large increase in the field for the microfluidic cell is purely (mostly) a simple function of a lower pathlength and $V=Ed$? If so this should be articulated.

8. "acted as an additional in series capacitance" Grammar

9. "the positive OEEF polarity (left panel in Fig. 4a) kinetically favours the most thermodynamically stable reaction path." I don't understand this sentence.

10.

Reviewer #3:

Remarks to the Author:

This is a very important demonstration of the role of oriented external electric field (OEEF) in driving molecular catalysis. The experimental work is solid and the metal free Huisgen reaction is elegantly demonstrated. The superiority over the conventional Cu(I) catalysts are also demonstrated clearly.

The work is excellent in light of the new metal free catalysis being shown here under the presence of electric field. The important previous computational prediction of the metal free Huisgen reaction under OEEF needs to be clearly mentioned in the context of this work (ref. 5).

We would like to thank you for your kind assistance in reviewing this work and thank the reviewers for their positive evaluation and, in particular, for their very constructive feedback which has pushed us to carry out additional experiments and significantly strengthen the manuscript. As a result of the additional experimental data, we have now included a new co-author, Roger Sanchis-Gual, who has actively participated in performing the new experiments.

Please find below our point-by-point detailed answers to each Referees' comments and queries together with indications of modifications made in the main manuscript and introduced additional data, accordingly. The original reviewer's comments are in *black italic* and our reply in **blue**.

REVIEWER COMMENTS:

Reviewer #1 (Remarks to the Author):

For the paper, titled "electrostatic catalysis of a click reaction in a microfluidic cell", Sevim et al. quantified the role of the electrical double layer on the electrostatic catalysis of a prototypical Huisgen cycloaddition with controllable mass transport. They designed a microfluidic cell, which could enhance reagent transport towards the electrified reactive interface and avoid both turbulent flow conditions and poor control of the convective mass transport. Their designed microfluidic reactor presents superior catalytic performance for cycloaddition reaction by 200% larger at a small, applied voltages of 2 V than that over a conventional Cu(I) chemical counterpart.

We thank Reviewer 1 for his/her positive highlights of our work and for the comments below which helped improving the quality of the manuscript.

Here are some comments for the paper to address before the consideration of publications.

- 1. What's the curvature of Au electrode? Fig 1 shows flat slab of Au. This is very confusing. How the Au electrode without any curvature can achieve such a high electric field of ~ 1 V/nm at a very small, applied potential?*

We indeed used flat Au electrode surfaces in our experiments. The schematics of our microfluidic cell, represented in the main manuscript (**Fig. 1**) and SI (**Fig. S1**), are an accurate picture of the real device. All experiments were performed in a parallel-plate electrodes configuration. The voltage applied between the two parallel electrodes creates an interfacial electrical double layer (EDL) in the vicinity of the electrodes' surfaces. The formation of such interfacial EDL is inherent to all electrified electrodes in contact with a polar liquid environment and the driving force for electrochemical redox reactions. The formation of such interfacial EDL capacitor results in the localization of an oriented external electric field (OEEF) near the electrodes' surface, as most of the applied voltage across the two electrodes drops within the several nanometers above the electrodes surfaces where the EDL capacitor fully develops.

- 2. Did authors explore a synergistic effect of combining both electrostatic reactions and Cu(I) catalysts? Since Cu is positively charged, will that further enhance the local Efield?*

Synergistic effects would definitely be very attractive towards enhancing the electrostatic catalysis. Unfortunately, the Cu^+ catalyst is a redox active specie which would immediately oxidize under the applied electrostatic conditions, rendering catalyst inactivation in this case and shutting down the chemical catalysis path.

- 3. The calculated OEEF magnitudes need a further validation from experiments, e.g., stark vibrational frequency.*

Measuring vibrational Stark effects inside our microfluidic cell under the operating conditions (*i.e.*, parallel-plate electrodes separated by a 250 μm -thick spacer) would be unreasonably challenging from a technical viewpoint. The employed chronoamperometry (CA) measurements have been used extensively in the past for such purpose (see measurements of differential capacitance of the interfacial EDL in *J. Phys. Chem. C* 112, 7486–7495, 2008 and *J. Phys. Chem. Lett.* 9, 126–131, 2018) and allow us assessing *in-situ* the OEEF magnitudes under the operations conditions in the microfluidic cell. The EDL capacitance values measured

here are in line with the vast previous electrochemistry work, and our new aim in this work is to correlate EDL capacitance with the performance of the studied electrostatic catalysis.

4. *A lack of information about the selectivity and conversion rates of the reaction when using OEEF. Will that be different as compared to the conventional Cu(I) catalyst?*

As a response to a similar query from reviewer 2 (comment 2), we have now included additional experiments to characterize the kinetics of the electrostatic catalytic process (see **Figure R3**, below), and compare reaction rates in both chemical and electrostatic catalysis paths. If by selectivity, the reviewer refers to regio and stereoselectivities, this has not been explored in this work.

5. *The author's description on page 8 only focuses on the positive potential. Could the authors elaborate on this observation at negative potentials? In general, if authors applied the field correctly, negative potential should show opposite trend for the reaction performance as compared to that under a positive electric field.*

We have discussed the effect of OEEF polarity in section: “*OEEF polarity effect on the electrostatic catalysis*”, which includes **Fig. 4** (as **new Fig. 5** in the revised manuscript). The results show indeed a contrary effect on the electrostatic catalysis with reaction yields significantly dropping 73% as compared to the positive polarity at the low applied voltage of -0.75 V. We have now expanded these results by bringing additional data at negative polarities for different voltages -0.75 V, -1.5 V and -2.0 V (**Figure R1** and **Figure R3**, below), which compares the outcome to the homologous positive voltages. While slight differences in the shape and magnitude are observed across the different voltametric signals due to possible batch-to-batch variations in the packing density of the surface functional groups, the voltage- and polarity dependence of the final catalytic yield is consistent across all samples. In particular, the positive and negative polarities obtained at the same applied bias of ± 0.75 V (**Figure R1a(i)**) and of ± 2.00 V (**Figure R1a(ii)**, **Figure R1b** and **Figure R3b**) come from the same batch and constitutes a direct comparison. We have now incorporated the additional data on the OEEF polarity effect (**Figures R1a(i-ii)**, **R1b(i)** and **R3b**) into the SI as a new supplementary figure (**Fig. S9**) and correspondingly called the figure in the main manuscript.

Figure R1. Cyclic voltammetry signals of electrostatic catalysis performed at different polarities. **a** compares the results obtained at (i) ± 0.75 V bias voltages; and (ii) ± 2.00 V bias voltages. **b** Results obtained from a different batch. (i) presents the results obtained at ± 2.00 V and -1.50 V, and (ii) shows the results from control groups together with the polarity dependence of electrostatic catalysis at ± 2 V applied bias.

6. On page 11, while discussing Fig. 4b, you attribute the effects of OEEF polarity to the stabilization of the charge-separated state in the transition state. Besides the provided references, are there any additional experimental or theoretical evidence that you can present to substantiate this stabilization effect in the transition state of the azide-alkyne cycloaddition reaction?

The TS stabilization under an OEEF is currently the most substantiated mechanisms supporting electrostatic catalysis. Such model is not exclusive to the azide-alkyne cycloaddition reaction but to all reactions bearing a degree of charge separation in the transition state. This hypothesis is extensively supported by numerous theoretical as well as experimental reports (see compendiums in references [4, 40, 41] from the main manuscript). As highlighted by Reviewer 3, the reference *Phys. Chem. Chem. Phys.* 19, 22482–22486, 2017 is a direct computational prediction of TS stabilization by OEEF in a metal free azide-alkyne click reaction.

7. More in-depth explanations of DFT calculations are needed, such as the model itself, how they added external field, how the TS was stabilized by the field, does the model solely incorporate azide-terminated molecules, or is gold (Au) also included as a catalyst? In addition, would variations of the reaction species potentially impact the change in dipole moment?

We have used DFT solely to get more accurate vectorial magnitudes of electrical dipoles for all isolated reactants and reaction products in the employed solvents (see Fig. S8 and Table S2 in the SI). The magnitudes of the dipole moments help us rationalize the reasons for the larger reaction yields in the positive OEEF polarity: (1) the magnitude of the reactants dipole moments (~ 1.5 D) times the electric fields generated within the microfluidic cell (~ 1 V/nm) yield energies well above kT , which results in effective reactants' orientation along their dipoles under the applied OEEF. (2) The reactants orientation for a positive OEEF polarity offers a more favorable approach for the formation of the transition state (TS)

between the Au-anchored terminal azide and the free Fc-alkyne (**Fig. 5a** in the revised manuscript), leading to the final 1,5-isomer product. The latter reaction scores the larger dipole moment increment (see **Table S2**), assuming similar dipole values for both TS and products given their structural homology (**Table S2** and **Fig. 5** in the revised manuscript). We have expanded this section to clarify all the above and to also account for Reviewer 2's queries on the used terminology.

8. *The manuscript would benefit from providing more detail on experimental reproducibility. Were the experiments performed multiple times, and were the results consistent? If the authors did multiple times, can they add the error bar for the results?*

We have now added errors bars to the new **Fig. 2b** and details of the statistics in the figure caption.

Reviewer #2 (Remarks to the Author):

In the attached manuscript, Sevim and colleagues present data regarding a microfluidic device that can perform electrostatic catalysis through a static field generated by a potential difference between two electrodes. They demonstrate catalysis of an azide-alkyne cycloaddition reaction without the exogenous need for Cu catalysts. On the whole, the work appears to be a nice contribution to the discipline of chemical catalysis, demonstrating how device engineering and electrostatics can be leveraged together to build out a new way to catalyze chemical reactions. This reviewer has not been active in this precise field for some time, and must confess that superficially the work appears similar to the decade-old pioneering studies by Gorin et al. It may behoove the authors to be clearer in their introduction and discussion what the differences/improvements are between their system and previously-reported ones. On the whole, the work is well argued and evidenced, and seems potentially worthy of publication.

We thank to Reviewer 2 for the positive general remarks about our work. We want to stress here our results bring several new key aspects over the previous pioneering work mentioned by the reviewer ([9,10] in the manuscript): (1) our work is an example of pure electrostatic catalysis in the absence of catalyst (neither homogeneous nor heterogeneous catalyst is present), (2) we show electrostatic catalytic effects at much lower applied potentials (<1V) and quantify the effect against the conventional Cu⁺ chemical catalysis, and (3), our microfluidic reactor allows us to perform these reactions under continuous (and laminar) flow conditions, which facilitates mass transport of reactants to the interfacial EDL where the electrostatic catalysis occurs and results in higher reaction yields. We have added the above points in **page 3** (see, revised manuscript) to emphasize the novelty points of this work.

The following are specific comments, major followed by minor:

1. *The authors refer in many places to their reaction occurring in a confined area/space. I'm not sure if I agree with this assessment. Generally, the term "confined" when used by supramolecular/polymer/colloid chemists refers to scenarios when movement is restricted to molecular length scales. The author's device is 250 μm tick, which is still nearly one million times larger than molecular dimensions. It seems to me that the molecule's diffusivity properties in the device would be quite similar to a bulk phase.*

The term "confined" in the manuscript refers exclusively to the interfacial EDL space (few nanometers above the electrodes surface) where the OEEF builds up and the electrostatic catalysis takes place. We have emphasized this in page 3 of the manuscript.

As per the molecular diffusivity in our microfluidic channel, **Figure R2**, which is adapted from one of our recent works (*Advanced Materials* 33 (30), 2101777, 2021), shows numerical mass transport simulations demonstrating that the concentration profile achieved in a very similar microfluidic device is drastically different than that in the bulk case (*i.e.*, droplet), even though the diffusion coefficient of the NH₃ in DMSO is same for both cases (see cases (i) and (ii), respectively for bulk and confined conditions). Basically, the concentration front is vertically moving in the bulk case (**Figure R2d(i)**), whereas its movement is in horizontal direction in the microfluidic device (**Figure R2d(ii)**) due to the suppressed convective motion of fluid elements in the confined case (see the fluid velocity maps in **Figure R2c**).

Figure R2. **a.** Illustration showing the confined reaction space in the microfluidic device. **b.** Representation of the (i) droplet (bulk), (ii) microfluidic (confined), and (iii) hypothetical (non-confined) cases considered in the numerical analyses. **c.** Velocity maps after 30 min of simulation time for (i) droplet (bulk), (ii) microfluidic (confined), and (iii) hypothetical (non-confined) cases. **d.** Concentration maps after 30 min of simulation time for (i) droplet (bulk), (ii) microfluidic (confined), and (iii) hypothetical (non-confined) cases. In panel d (iii) lines of equal concentration shown in back, representing the orientation of the concentration front.

2. The authors are keen to make the claim that their device catalyzes the cycloaddition reaction, and based on the data in Fig. 2a, it seems that this is likely the case; however, catalysis has to do with the acceleration of rates, and the authors do not directly report this because (as far as I can tell) all reactions were conducted for 30 min without any direct measurement of kinetics. Minimally, the authors would need to report conversions as a function of time to obtain rates, and then OEEF-induced catalysis could then be more solidly supported based on rate accelerations.

We agree with Reviewer 2 that a kinetics study would reinforce the results of our work. We have now performed time-dependent experiments and build corresponding kinetic curves (**Figure R3**). **Figure R3d-f** has been now added to the main manuscript (as a **new Fig. 3**) together with experimental detail and corresponding discussion (see revised manuscript **page 8-9**). Overall, we need twice as much reaction time in the chemical catalysis to reach saturation of Fc surface coverage on the electrode as compared to the electrostatic catalysis at 2 V. The average slopes (average reaction rates expressed as 2.03×10^{-2} and 5.24×10^{-2} Fc/nm² min⁻¹ respectively for Cu⁺ and electrostatic catalysis cases) correspondingly yield 2.58 times as much conversion rates for the electrostatic case. While carrying out the new experiment to answer the reviewer's queries, we collected additional SI which adds reproducibility to the polarity dependence of the electrostatic catalysis (**Fig. R3b**) as well as completes the SI data with more control experiments (**Fig. R2a-c**) which has been now added to a new SI Fig. S9. The results are fully consistent with the rest of the presented data.

Figure R3. a. Multiple CV cycles at different scan rates (SR), such as 0.1 V s⁻¹ (black), 0.2 V s⁻¹ (red), 0.3 V s⁻¹ (blue) performed on the sample prepared with electrostatic catalysis at a bias voltage of 2 V. b. CVs obtained from the samples prepared with electrostatic catalysis at 2 V bias voltage for 30 min respectively +2 V (red) and -2 V (blue) together with control experiments performed for 60 min in the absence of both electric field ($V_{\text{applied}} = 0$ V) and Cu⁺ catalysis (grey), together with the CVs obtained from bare functionalized (green) and blank Au (orange) electrodes. c. First- and last-five CV scans showing the surface contaminations during Cu⁺-based chemical catalysis. d. CVs obtained from the samples prepared with electrostatic catalysis at a bias voltage of 2 V for 10 min (black), 20 min (red), 30 min (blue), and 60 min (green) to compare with the control experiment for 60 min (grey). Note that CVs are obtained from different samples prepared in separate and independent experiments. e. CVs obtained from the samples prepared with Cu⁺ catalysis for 10 min (black), 20 min (red), 120 min (green) to compare with the samples prepared with electrostatic catalysis for 30 min (dashed blue) and 60 min (dashed grey). f. Reaction yield as a function of time for electrostatic catalysis (blue solid line with circles) and Cu⁺ catalysis (green solid line with circles), and their corresponding linear fits in color coded dashed lines.

3. *Reaction progress is monitored through the current density associated with the redox couple of Fc/Fc+, and using this assay the authors can conclude that higher fields result in more surface-attachment of ferrocene. It seems like it would behoove the authors if this measure of yield could be converted to a more conventional measurement (such as % of azides on surface that got converted). This could be done by establishing a calibration curve constructed by the in solution, putting it down on the Au surface, and measuring the current density when a known amount of ferrocene is down on the surface*

We have now converted the anodic charge density to the ferrocene coverage (*i.e.*, the number of ferrocene per nm²) in the kinetic study (new **Fig. 3c** and **Fig. R3f**) and benchmarked this with the coverage of an ideal fully covered ferrocene layer, which have been extensively reported in the literature (*J. Am. Chem. Soc.* 112, 4301–4306, 1990 and *Langmuir* 18, 1288–1293, 2002). Moreover, we have also updated **Fig. 2b** accordingly.

4. *The following sentence in the manuscript seems quite incorrect: “The OEEF polarity impact on the reaction is 2-fold; (i) on the kinetics; it increases reaction yield of the azide-alkyne cycloaddition by facilitating the proper alignment of the free ethynylferrocene reactant in solution with respect to the surface-bound azide group³⁶ (Fig. 4a), and (ii) on the thermodynamics; it stabilizes the charge separated state of one of the resonance contributors in the transition state (TS) (Fig. 4b), thus reducing the energy barrier.”*

Firstly, kinetics is about rates – not yields. Returning to my point #2, the authors ought to make more direct measurements on rates to make claims about kinetics. The second part of the sentence does not seem correct either. Thermodynamics governs which species prevail at equilibrium. Stabilizing the charge configuration of a transition state would accelerate a reaction and impact kinetics, not thermodynamics.

- A. *In connection to this point, since the azide-alkyne cycloaddition is effectively irreversible, it is under kinetic control, not thermodynamic control. So what matters in controlling the rate is the change in dipole moment upon going to the transition state: $\Delta G^{\ddagger} = \vec{E} \cdot \Delta \vec{\mu}_{R \rightarrow TS}$.*

We agree with the reviewer that our original arguments to explain the different aspects of reaction acceleration were not adequately selected. We have also covered the kinetics study also mentioned in previous comment and new results now included in the **Fig. R3** and **new Fig 3** in the revised manuscript. We have also modified the text in this section accordingly (pages 12-13) to account for the new kinetic study as well as the new DFT calculations of dipole moments done in response to reviewer 1's queries, including the dipole change analysis mentioned by this reviewer.

5. *The last part of the paper shows some interesting experiments looking at the effect of solvent polarity and showing that the effect on yield is consistent with the predicted decrease in the EDL magnitude. Could this be compared to classic electrostatic theory? Also, would the authors predict a further acceleration if the linker of the alkane thiol were shorter? Presumably 11 carbons would also place the azide further away from the strongest portion of the EDL?*

We think this is an excellent point. A systematic study of the effect of the position of the click reactants within the EDL in the electrostatic catalysis would indeed be an interesting study but will require a long study scrutinizing different lengths for the thiol linkers. While this is outside the scope of this work, we have now conducted a preliminary test using a shorter alkyne-terminated linker to the Au electrode (**Figure R4a-b**), which now reacts with a 0.1M ferrocenoyl azide in acetonitrile in the microfluidic device for 30 minutes at an applied OEEF of 1 V. The comparison to the homologous Cu⁺ based catalysis under the same conditions in the absence of an OEEF suggests superior catalytic efficiency compared to the longer linker (**Fig. 2b** in the main manuscript), where at such moderate OEEFs, the reaction yields were almost equivalent. While reactions in **Figure R4a** and **b** are substantially different and would not be an accurate comparison in the current study, we thought the observed preliminary results in **Figure R4** should be included here, and that they point to a future interesting study enabling probing the electrostatic profile across the EDL.

Figure R4. a-b. Different molecules utilized to perform click reaction. c CVs obtained from the samples prepared with the molecules presented in panel b via electrostatic catalysis at a bias voltage of 1 V for 30 min (blue), and via Cu^+ catalysis (green) to compare with the control experiment (red) performed in the absence of both $\text{Cu}(\text{I})$ and electric field for 30 min

1. *“Following nature’s inspiration, the above catalytic processes is conducted under controlled mass transport conditions via a confined microfluidic channel, where turbulent flow and radial forces are avoided.” If by Nature, the authors are referring to biology, then this sentence does not make sense because biology does not use microfluidic channels to promote catalysis (as far as I know)*

We referred here to key biological processes such as enzymatic catalysis where reactions take place in confined reactors (nanometre size enzyme active sites) with unconventional mass transport and where electric fields have been suggested to be a key effector for enzymatic catalysis (see refs [1,3,6] from the manuscript). We have removed the indicated mention from the text to avoid confusions and leave the general bioinspirational concepts already mentioned in the introduction section of the manuscript.

2. *“We believe this is a significant jump towards the implementation of a clean electrostatic catalysis in chemical industry applications.” Seems like rather editorialized language*

We have rephrased this sentence in the revised manuscript (**page 3**).

3. *“Upon reaction is completed.” Grammar*

We have corrected the sentence in **page 3** of the revised manuscript.

4. *Showing the absence of reaction in the absence of field in Figure 2 would be helpful (though it looks like the authors did this experiment and reported it in Figure S6)*

This is now included in **new Figure 3a** and supplementary CV graphs in the SI (**new Fig. S9**) showing the absence of reaction in the absence of electric field (see **Figure R3b,d**).

5. *In Table 1, the authors calculate the electric field based on a RC model. It would be good to show the equation of how this was derived based on the observed quantities. Also would be good to have error bars in Table 1.*

The corresponding equations and the derivation of calculated quantities from the measured ones are in the SI section “*Supplementary Note 4: Electrochemical characterization*” (see SI **pages 14-16**). We have now added the error bars in **Table 1** and **Figure 4c** of the revised manuscript.

6. *Related to the previous point, if I'm not mistaken there is pretty good analytical theory for predicting EDL fields like Chapman-Gouy? Couldn't the authors compare their values to that from these classic models?*

Our interfacial EDL arises from the alignment of polar acetonitrile molecules near the electrode surface and does not consider solvated ions as proposed by the Guy-Chapman theory.

7. *The large increase in the field for the microfluidic cell is purely (mostly) a simple function of a lower pathlength and $V=Ed$? If so this should be articulated.*

Although the formation of boosted OEEF is mainly based on the EDL generated in the polar acetonitrile solvent, in the case of microfluidic cell (where electrodes are separated by 250 μm , in contrast to the conventional electrochemical cell where electrodes separated by 7 mm), we have calculated larger OEEF magnitudes. While we don't have a definitive explanation for this phenomenon, we hypothesize that the confinement of the liquid between the two electrodes in the microfluidic cell results in a more efficient alignment of acetonitrile molecules yielding a more compact EDL and a higher OEEFs. We have clarified this in **page 12** of the revised manuscript.

8. *“acted as an additional in series capacitance” Grammar*

We have corrected the sentence (**page 12**).

9. *“the positive OEEF polarity (left panel in Fig. 4a) kinetically favours the most thermodynamically stable reaction path.” I don't understand this sentence.*

This section has been fully revisited in response to previous reviewers; 1 and 2 queries.

Reviewer #3 (Remarks to the Author):

This is a very important demonstration of the role of oriented external electric field (OEEF) in driving molecular catalysis. The experimental work is solid and the metal free Huisgen reaction is elegantly demonstrated. The superiority over the conventional Cu(I) catalysts are also demonstrated clearly.

The work is excellent in light of the new metal free catalysis being shown here under the presence of electric field. The important previous computational prediction of the metal free Huisgen reaction under OEEF needs to be clearly mentioned in the context of this work (ref. 5).

We appreciate the supportive feedback. Following the referee's suggestion, we have now clarified the importance of previous computational works mentioned in the introduction in **page 2** of the revised manuscript.

Reviewers' Comments:

Reviewer #1:

Remarks to the Author:

The authors have addressed all my comments.

Reviewer #3:

Remarks to the Author:

The excellent work has been suitably revised. Strongly accepted for publication

Reviewer #4:

Remarks to the Author:

The authors have adequately addressed most of the raised points. They have provided conversions over time, offering a direct correlation to reaction rates. Additionally, they have successfully converted the anodic charge density into ferrocene coverage. However, the authors need to provide commentary on the percentage of azides that underwent conversion, as a calibration curve for this aspect is currently absent. The authors should explain why such a calibration curve has not been included at this stage.

Although the manuscript now includes a comparative study on linker chain length, it's important to note that these involve two distinct reactions. It remains uncertain what this comparison contributes to the overall story of the manuscript. To enhance the manuscript's quality, I propose that the authors consider comparing the same reaction by shortening the linker chain length. This addition would substantially improve the manuscript and provide more meaningful insights. I strongly recommend the authors include such a study in their revised manuscript.

We are glad to see the major reviewers' concerns have now been resolved. Reviewer 4 raises interesting points which, while we think are beyond the scope of the present work, we have noted and included into our follow-up work currently under course in our lab.

Below our point-by-point detailed answers to each Referees' comments. The original reviewer's comments are in *black italic* and our reply in **blue**.

REVIEWER COMMENTS:

Reviewer #1 (Remarks to the Author):

The authors have addressed all my comments.

We thank Reviewer #1 for his/her positive feedback.

Reviewer #3 (Remarks to the Author):

The excellent work has been suitably revised. Strongly accepted for publication

We appreciate Reviewer #3's positive highlights and feedback.

Reviewer #4 (Remarks to the Author):

The authors have adequately addressed most of the raised points. They have provided conversions over time, offering a direct correlation to reaction rates. Additionally, they have successfully converted the anodic charge density into ferrocene coverage. However, the authors need to provide commentary on the percentage of azides that underwent conversion, as a calibration curve for this aspect is currently absent. The authors should explain why such a calibration curve has not been included at this stage.

We are glad Reviewer #4 acknowledges the main points of concern has been already addressed. A quantification of converted azides could be directly inferred from the ferrocene coverage represented for instance in Fig. 2b. Our benchmark calibration here is the coverage of an ideal fully covered ferrocene layer (page 5 of the main manuscript), which has been extensively reported in previous literature (J. Am. Chem. Soc. 112, 4301–4306, 1990; Langmuir 18, 1288–1293, 2002) and in more recent revisiting (*Langmuir*, 38, 11, 3585–3596, 2022). An additional caveat in such quantification would be that the packing density of the azide-terminated molecular layer would likely be higher than the packing density of the ferrocene terminated ones due to size mismatch between the two end groups, which would make difficult an accurate quantification of converted azides. Fig. 2b show the surface of the electrode already reaches saturation at 2V, and that additional ferrocene cannot be chemically attached, *i.e.*, no further azide conversion is possible. The purpose of designing a surface reaction was indeed to quantify the effect of the OEEF on the click reaction, which means the relevant plots are Fig. 2b and the new 3c. Fig. 2c represents here our calibration curve, where a particular ferrocene coverage (or estimated azide conversion) can be achieved for a given applied OEEF. The chemical analysis aspect is beyond the aim of this work, but it is a very interesting point for our next follow-up work, where reactants will be converted at the electrode surface and the reaction products collected at the end of the microfluidic channel for more conventional chemical analysis. Thank you.

Although the manuscript now includes a comparative study on linker chain length, it's important to note that these involve two distinct reactions. It remains uncertain what this comparison contributes to the overall story of the manuscript. To enhance the manuscript's quality, I propose that the authors consider comparing the same reaction

by shortening the linker chain length. This addition would substantially improve the manuscript and provide more meaningful insights. I strongly recommend the authors include such a study in their revised manuscript.

We want to clarify here that this work will not include the mentioned comparative study, as this was preliminary data we exclusively shared to answer one of previous reviewer's queries. These preliminary results provide an insight and open to a very interesting study to quantify the EEF profiles near the electrode, as this reviewer also infers, but they are currently insufficient to draw a solid conclusion and will require a long systematic study examining different lengths for the thiol linkers and using the same reaction as suggested by the reviewer. We are now very excited to follow up this direction in future work.